# Protocol for a scoping review of time to treatment in adults with newly diagnosed multiple myeloma

Matthew R. LeBlanc[1,2*], Allison O. Taylor[3], Osborn Owusu Ansah[4], Jamie Conklin[5]

1 School of Nursing, University of North Carolina at Chapel Hill, Chapel Hill, North Carolina, United States of America, 2 Lineberger Comprehensive Cancer Center, University of North Carolina at Chapel Hill, Chapel Hill, North Carolina, United States of America, 3 Department of Medicine, Division of Hematologic Malignancies and Cellular Therapy, Duke University, Durham, North Carolina, United States of America, 4 School of Nursing, Duke University, Durham, North Carolina, United States of America, 5 Health Sciences Library, University of North Carolina at Chapel Hill, Chapel Hill, North Carolina, United States of America

* matthew_leblanc@unc.edu

## Abstract

### Background

Delays in cancer treatment can result in tumor growth, increased clonal heterogeneity, upstaging, increased symptoms, organ damage, increased psychological distress, and worse clinical outcomes. Evidence supported guidelines for treatment timeliness exist in many cancers, but not for multiple myeloma (MM) though there is reason to believe delays in treatment would be detrimental.

### Aims

This scoping review aims to explore what is known about the impacts of the time from diagnosis to treatment among patients with MM.

### Methods

Our review will be guided by the Joanna Briggs Institute scoping review methodology. Our search strategy was developed to identify sources published in or after the year 2000 related to time to treatment for adults with newly diagnosed multiple myeloma in PubMed, CINAHL, Scopus and Embase. Sources will be screened by two independent reviewers after exclusion/inclusion criteria are pilot tested and refined. A data extraction form was developed and will be refined by the study team for use during the scoping review.

### Discussion

This review will summarize the landscape of research related to time to treatment among newly diagnosed adults with multiple myeloma. Raising awareness of the available evidence on this topic, within the MM clinical and research community, will guide future research to address identified knowledge gaps.

**Data availability statement:** No datasets were generated or analysed during the current study.

**Funding:** The author(s) received no specific funding for this work.

**Competing interests:** The authors have declared that no competing interests exist.

## Registration

This protocol was registered with Open Science Framework (OSF) on the 25th of March, 2025 and can be found at **osf.io/sydqw**.

## Introduction

Delays in cancer treatment can result in tumor growth, increased clonal heterogeneity, upstaging, increased symptoms, organ damage, increased psychological distress, and worse clinical outcomes including increased mortality [1–3]. Khorana and colleagues found an increase in absolute risk of mortality between 1.2–3.2% per additional week of treatment delay across patients with breast, prostate, colorectal, non-small cell lung cancer, renal and pancreatic cancer [1].

For many cancers there is enough evidence to support clinical care recommendations for the timeliness of treatment initiation (TTI). These include, breast, colon, esophageal, head and neck, non-small cell lung, and melanoma. Guidelines like this do not exist for MM, though there is reason to believe that timeliness is important to clinical outcomes in MM. Delayed diagnosis and treatment are associated with increased symptom burden and worse disease-free survival [4–7]. Delayed diagnoses in MM, leading to delayed initial treatment, has been shown to increase disease related symptoms, cause end organ damage and worsen progression free survival [5].

Our larger goal is to generate evidence to inform clinical care guidelines around the optimal timing of treatment initiation in multiple myeloma. The purpose of this review is to survey the landscape of TTI evidence for those with newly diagnosed multiple myeloma, including the impacts of TTI on clinical outcomes. A preliminary search of MEDLINE, the Cochrane Database of Systematic Reviews and *JBI Evidence Synthesis* was conducted and no current or underway systematic reviews or scoping reviews on the topic were identified.

## Methods

This scoping review aims to explore what is known about the impacts of the time from diagnosis to treatment among patients with MM. A scoping review was selected because it is an appropriate, systematic method for mapping what is known and what is yet to be explored in an area of interest. This scoping review will be conducted in accordance with the JBI methodology for scoping reviews [8]. This protocol was registered with Open Science Framework (OSF) March 25, 2025 and can be found at **osf.io/sydqw**. At the time of this protocol manuscript submission, our team has developed our search strategy and completed our screening pilot as described below. We anticipate to complete screening in 2 months time, data extraction in 3 months and to report results in 4–5 months. Our PRISMA-P ((Preferred Reporting Items for Systematic review and Meta-Analysis Protocols) checklist can be found in Appendix I in S1 File [9].

## Review question

The purpose for conducting this scoping review is to map the evidence as it relates to time from diagnosis to treatment among adults living with MM. Our sub questions include:

1. Identify definitions and measures of TTI in MM.

2. Characterize reported time frames from diagnosis to treatment initiation.

3. Explore factors associated with delays in treatment.

4. Assess the impact of TTI on clinical outcomes (OS, PFS, symptom burden).

5. Identify gaps for future research.

## Types of sources

This scoping review will consider experimental and quasi-experimental study designs (e.g., randomized controlled trials, non-randomized controlled trials, interrupted time-series studies). Observational studies including prospective and retrospective cohort studies, case-control studies and cross-sectional studies, and descriptive observational study designs will be considered for inclusion. Systematic reviews, commentaries and opinion papers will also be considered for inclusion in this scoping review.

## Search strategy

An initial limited search in PubMed was undertaken to identify articles on the time to treatment in MM. Terms used in the titles and abstracts of relevant articles, and the index terms used to describe the articles were used to develop a full search strategy for PubMed under the guidance of a Health Sciences librarian (JC) (see Appendix II in S1 File).

Our search strategy was guided by the concepts identified in Table 1. Studies published in any language will be included where English translations are available. If English translations are not available these sources will be excluded. Studies published in or after the year 2000 will be included. Treatments for MM before 2000 were only moderately effective, and highly toxic dampening enthusiasm for early treatment. Since then, the treatment landscape has shifted dramatically, with more than 30 FDA approvals [10]. The increasing effectiveness and tolerability of new treatments have changed the costs and benefits of earlier treatment dramatically. Now there is a strong rationale and an increased appetite for earlier treatment which led to our decision to limit the search in this way.

To validate our search strategy, we a priori identified four sources that a search strategy must identify to successfully achieve our aim of uncovering sources related to time to treatment among adults with newly diagnosed multiple myeloma, and then made sure our search strategy did in fact identify these sources. After validation of our search strategy, including all identified keywords and index terms, it was adapted for CINAHL Plus with Full Text (EBSCO*host*), Scopus and Embase (Elsevier).

**Table 1. Frame for review question and important literature search concepts.**

| Population | Adults (age 18 +) with newly diagnosed with multiple myeloma |
|---|---|
| Concept | Time from multiple myeloma diagnosis to treatment initiation (including predictors of and effects of.) |
| Context | Sources published in, or translated into, English on or after the year 2000. |

## Study/source of evidence selection

Following the search, identified citations will be uploaded into EndNote *X9 (Clarivate Analytics, PA, USA)* and duplicates removed. De-Duplicated citations will be uploaded into Covidence systematic review software *(Veritas Health Innovation, Melbourne, Australia)*.

Screening will occur following a screening pilot test in which we will screen 25 sources and refine and modify our initial inclusion/exclusion criteria based on our search strategy concepts (Table 1). Reviewers will not be blinded during the screening process. Titles and abstracts will then be screened by two independent reviewers for inclusion in the review. Full text of sources identified for further review will be assessed independently by two reviewers for final inclusion. Reasons for exclusion of sources during full text review will be recorded and reported. After full text sources have been selected for inclusion, we will screen sources in their reference lists and also screen sources that have cited these included sources (identified in SCOPUS) using the procedures outlined above. Any disagreements that arise between reviewers at each stage of the selection process will be resolved through discussion until consensus.

The results of the search and the study inclusion process will be reported in full in the final scoping review and presented in a Preferred Reporting Items for Systematic Reviews and Meta-analyses extension for scoping review (PRISMA-ScR) flow diagram [11].

## Data extraction

Data will be extracted from papers included in the scoping review by two or more independent reviewers using a data extraction tool developed by the reviewers. The data extracted will include specific details about the participants, concept, context, study methods and key findings relevant to the review question/s.

A draft extraction form is provided (see Appendix III in S1 File). The draft data extraction tool will be modified and revised as necessary during the process of extracting data from each included evidence source. Modifications will be detailed in the scoping review. Any disagreements that arise between the reviewers will be resolved through discussion. If appropriate, authors of papers will be contacted to request missing or additional data, where required.

## Data analysis and presentation

Data will be summarized narratively and organized around answering our scoping review questions. We will summarize the ways time to treatment is defined and measured in the literature. We will describe narratively how sources defined and/or conceptualized time to treatment. We will characterize reported time frames of time to treatment in myeloma. Results will be stratified by transplant eligibility and induction treatment regimen as the data allows. We will describe factors associated with delays in treatment identified. We will summarize the ways the impact of time to treatment on clinical outcomes (survival, symptom burden) has been explored, and we will identify gaps in our knowledge on time to treatment in myeloma to guide future research.

## Discussion

This scoping review is part of a larger program of research exploring access to high quality multiple myeloma care, including generating evidence to inform clinical care guidelines on the timeliness of treatment initiation. Clear standards and guidelines will allow us to evaluate the quality of current clinical care and provide the evidentiary support to encourage clinical practice changes. This review builds the foundations for our future work by summarizing the current landscape of evidence related to time to treatment among newly diagnosed adults with multiple myeloma. Understanding the available evidence and current knowledge gaps on this topic will guide future research.

## Supporting information

**S1 File.   Appendix I.** PRISMA-P (Preferred Reporting Items for Systematic review and Meta-Analysis Protocols). **Appendix II.** Search strategy. **Appendix III.** Data extraction instrument.
(DOCX)

## Author contributions

**Conceptualization:** Matthew R. LeBlanc, Allison O. Taylor, Osborn Owusu Ansah.

**Methodology:** Matthew R. LeBlanc, Jamie Conklin.

**Writing – original draft:** Matthew R. LeBlanc.

**Writing – review & editing:** Allison O. Taylor, Osborn Owusu Ansah, Jamie Conklin.

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
