## [Decision Letter · Decision Letter 0]

19 Jun 2025

Dear Dr. LeBlanc,

Thank you for submitting your manuscript to PLOS ONE. After careful consideration, we feel that it has merit but does not fully meet PLOS ONE’s publication criteria as it currently stands. Therefore, we invite you to submit a revised version of the manuscript that addresses the points raised during the review process.

**ACADEMIC EDITOR:**

Overall the manuscript is well written and potentially of great interest to the audience; however, some minor issues have to be addressed, as suggested by the Reviewers. 

Please revise the manuscript according to all Reviewers' suggestions and comments, in order to make it suitable for publication. 

Of particular interest would be to stratify the population into 2 groups: transplant-eligible and non-eligible patients (in order to provide more reliable information on OS and PFS), and according to treatment regimen applied due to expected differences in efficacy and prognosis; moreover, include detailed information about how differences in definition of TTI are managed during the analysis. 

We look forward to receiving your revised manuscript.

Kind regards,

Carmelo Caldarella, Ph.D., M.D.

Academic Editor

PLOS ONE

Journal Requirements:

**Additional Editor Comments:**

Dear Authors, I have read carefully and with interest your manuscript and evaluated Reviewers suggestions. Overall the manuscript is well written and potentially of great interest to the audience; however, some minor issues have to be addressed, as suggested by the Reviewers.

Please revise the manuscript according to all Reviewers' suggestions and comments, in order to make it suitable for publication.

Of particular interest would be to stratify the population into 2 groups: transplant-eligible and non-eligible patients (in order to provide more reliable information on OS and PFS), and according to treatment regimen applied due to expected differences in efficacy and prognosis; moreover, include detailed information about how differences in definition of TTI are managed during the analysis.

Best regards

Carmelo Caldarella, PhD MD

Editor

Reviewers' comments:

Reviewer's Responses to Questions

**Comments to the Author**

1. Does the manuscript provide a valid rationale for the proposed study, with clearly identified and justified research questions?

Reviewer #1: Yes

Reviewer #2: Yes

Reviewer #3: Yes

Reviewer #4: Yes

2. Is the protocol technically sound and planned in a manner that will lead to a meaningful outcome and allow testing the stated hypotheses?

Reviewer #1: Yes

Reviewer #2: Yes

Reviewer #3: Yes

Reviewer #4: Yes

3. Is the methodology feasible and described in sufficient detail to allow the work to be replicable?

Reviewer #1: Yes

Reviewer #2: Yes

Reviewer #3: Yes

Reviewer #4: Yes

4. Have the authors described where all data underlying the findings will be made available when the study is complete?

Reviewer #1: No

Reviewer #2: Yes

Reviewer #3: Yes

Reviewer #4: Yes

5. Is the manuscript presented in an intelligible fashion and written in standard English?

Reviewer #1: Yes

Reviewer #2: Yes

Reviewer #3: Yes

Reviewer #4: Yes

You may also provide optional suggestions and comments to authors that they might find helpful in planning their study.

Reviewer #1: Thank you for the opportunity to review this protocol. While I am not an expert in scoping review methodology, the manuscript appears to be well-structured, clearly written, and methodologically sound. The authors provide a clear rationale for the study and outline their objectives and methods in accordance with standard guidance. The topic is relevant and timely, and the proposed approach seems appropriate to map the existing literature in this area. I do not have any major concerns at this time.

Reviewer #2: Given the well-documented differences in outcomes between transplant-eligible and non-eligible patients, I recommend stratifying the study population into at least two groups to evaluate overall survival (OS) and progression-free survival (PFS), or alternatively focusing the analysis on just one of these categories. Therapeutic efficacy is not uniform across all regimens. Although IMiDs have been part of treatment protocols since 1999, their combination with proteasome inhibitors (PIs) became common only later in clinical practice. More recently, the introduction of anti-CD38 monoclonal antibodies has further impacted OS and PFS. If feasible, the data should also be stratified by treatment regimen—for example: VTD, VRD, RD, VD, VMP vs daratumumab-based or other monoclonal antibody-based regimens vs others.

Reviewer #3: This manuscript presents a scoping review protocol that aims to explore the evidence on time to treatment initiation (TTI) in adults newly diagnosed with multiple myeloma. The review is structured according to Joanna Briggs Institute methodology and PRISMA-ScR standards, with the goal of mapping definitions, outcomes, and research gaps related to TTI. This is a commendable and much-needed effort that aligns well with current research and clinical priorities in MM. My comments are given below;

Major:

1. Please clarify how differences in definitions of TTI will be managed during analysis. Will you categorize or synthesize them descriptively?

2. Consider discussing how language restrictions will be handled if English translations are not available.

3. Expand on how you plan to address publication bias and the potential variability in study quality.

4. Include a brief discussion of the anticipated contribution of this scoping review to clinical practice guidelines or research agendas.

5. Provide further rationale for the year 2000 as the search cut-off, beyond the general availability of newer treatments.

Minor:

1. Edit the abstract for clarity—avoid phrases like 'surgery' for describing transplantation.

2. In the methods, consider explaining whether reviewers will be blinded during abstract screening.

3. There is a duplicated reference (LeBlanc 2022, R6-8) that should be consolidated.

4. Include the final version of your PRISMA-ScR checklist as an appendix.

5. Table 1 could benefit from improved formatting for clarity in separating concepts and terms.

Suggested References:

1. Khorana AA, et al. Time to initial cancer treatment in the United States and association with survival over time: An observational study. PLoS One. 2019;14(3):e0213209.

2. Cone EB, et al. Assessment of Time-to-Treatment Initiation and Survival in a Cohort of Patients With Common Cancers. JAMA Network Open. 2020;3(12):e2030072.

3. Friese CR, et al. Diagnostic delay and complications for older adults with multiple myeloma. Leuk Lymphoma. 2009;50(3):392-400.

4. Kariyawasan CC, et al. Multiple myeloma: causes and consequences of delay in diagnosis. QJM. 2007;100(10):635-40.

5. LeBlanc MR, et al. A cross-sectional observational study of health-related quality of life in adults with multiple myeloma. Support Care Cancer. 2022;30(6):5239-48.

6. Peters MDJ, et al. Best practice guidance and reporting items for the development of scoping review protocols. JBI Evid Synth. 2022;20(4):953-68.

7. Tricco AC, et al. PRISMA Extension for Scoping Reviews (PRISMA-ScR): Checklist and Explanation. Ann Intern Med. 2018;169(7):467-73.

Reviewer #4: A well-designed and purposeful article. Raising awareness of the available evidence on this topic, within the MM clinical and research community, will guide future research to address identified knowledge gaps.

**Do you want your identity to be public for this peer review?** For information about this choice, including consent withdrawal, please see our Privacy Policy

Reviewer #1: No

Reviewer #2: No

Reviewer #3: No

Reviewer #4: No

---

## [Author Response · Author response to Decision Letter 1]

11 Jul 2025

Carmelo Caldarella, Ph.D., M.D.

RE: Manuscript ID PONE-D-25-22988

Greeting Dr. Caldarella,

Thank you to the editors and the reviewers for the thoughtful feedback on our scoping review titled ‘Protocol for a scoping review of time to treatment in adults with newly diagnosed multiple myeloma’. Thank you also, for allowing us the opportunity to improve our manuscript and resubmit. Below is a summary of reviewers’ comments and our responses. Please note that reviewers 1 and 4 raised no concerns and are therefore not represented in our summary. We feel that we have responded appropriately to reviewers’ concerns and that our manuscript has improved as a result. Thank you for your consideration.

Matthew R. LeBlanc PhD, BSN

Assistant Professor

School of Nursing

University of North Carolina at Chapel Hill

Reviewer 2 Comments

1. Given the well-documented differences in outcomes between transplant-eligible and non-eligible patients, I recommend stratifying the study population into at least two groups to evaluate overall survival (OS) and progression-free survival (PFS) //Thank you for this wonderful suggestion. We have added language to our analysis section that indicates that we will stratify results by transplant eligibility as the data allows. (page 6)

2. If feasible, the data should also be stratified by treatment regimen—for example: VTD, VRD, RD, VD, VMP vs daratumumab-based or other monoclonal antibody-based regimens vs others // We have added language to our analysis section indicating that we will stratify results by induction regimen as data allows.

Reviewer 3 Comments

1. Clarify how differences in definitions of TTI will be managed during analysis. Will you categorize or synthesize them descriptively? // Definitions will be explored descriptively. Results will not be combined as in a meta-analysis. This clarification is on page 6 (Data Analysis and Presentation).

2. Discuss how language restrictions will be handled if English translations are not available. // We do not have the capacity to translate sources and therefore these sources without English translations will be excluded. (See page 4 for added language).

3. Expand on how you plan to address publication bias and the potential variability in study quality. // We do not plan on assessing for bias or study quality. Not assessing for bias/quality is common in scoping reviews and would be quite tricky in our case given the mix of study types we plan on including.

4. Include a brief discussion of the anticipated contribution of this scoping review to clinical practice guidelines or research agendas. //Thank you for the invitation to clarify the impact we intent this work to have. We have added language to the Background (page 3) and to the Discussion (page 6)

5. Provide further rationale for the year 2000 as the search cut-off, beyond the general availability of newer treatments. // Thank you for the encouragement to clarify our choice here. We have added language to page 4 under ‘Search Strategy’ highlighting how changes in the treatment landscape have changed the cost/benefit of early treatment greatly increasing the rationale and appetite for earlier treatment.

6. Edit the abstract for clarity—avoid phrases like 'surgery' for describing transplantation. // We appreciate the careful reading of our manuscript. In this case it appears there was some mistake. We do not use the word surgery in the abstract or body of our paper.

7. In the methods, consider explaining whether reviewers will be blinded during abstract screening. // We have added language (page 5), clarifying that reviewers will not be blinded during the screening process

8. There is a duplicated reference (LeBlanc 2022, R6-8) that should be consolidated. // Thank you for your close reading of our manuscript. This duplication has been removed.

9. Include the final version of your PRISMA-ScR checklist as an appendix. We have included an our PRISMA-P checklist as an appendix. (Appendix I)

10. Table 1 could benefit from improved formatting for clarity in separating concepts and terms. // Our table has been updated to clarify the concepts presented. (page 7)

---

## [Decision Letter · Decision Letter 1]

8 Aug 2025

Protocol for a scoping review of time to treatment in adults with newly diagnosed multiple myeloma.

PONE-D-25-22988R1

Dear Dr. LeBlanc,

We’re pleased to inform you that your manuscript has been judged scientifically suitable for publication and will be formally accepted for publication once it meets all outstanding technical requirements.

Kind regards,

Carmelo Caldarella, Ph.D., M.D.

Academic Editor

PLOS ONE

Additional Editor Comments (optional):

Dear Authors, thank you for having addressed all the concerns raised by the Reviewers and for having rendered this manuscript more scientifically sound. The manuscript is now suitable for publication in this journal.

Best regards,

Carmelo Caldarella

Reviewers' comments:

Reviewer's Responses to Questions

**Comments to the Author**

1. Does the manuscript provide a valid rationale for the proposed study, with clearly identified and justified research questions?

Reviewer #1: Yes

2. Is the protocol technically sound and planned in a manner that will lead to a meaningful outcome and allow testing the stated hypotheses?

Reviewer #1: Yes

3. Is the methodology feasible and described in sufficient detail to allow the work to be replicable?

Reviewer #1: Yes

4. Have the authors described where all data underlying the findings will be made available when the study is complete?

Reviewer #1: Yes

5. Is the manuscript presented in an intelligible fashion and written in standard English?

Reviewer #1: Yes

You may also provide optional suggestions and comments to authors that they might find helpful in planning their study.

Reviewer #1: The revised manuscript shows significant improvement in both clarity and scientific content. I do not have any additional comments or suggestions at this time.

**Do you want your identity to be public for this peer review?** For information about this choice, including consent withdrawal, please see our Privacy Policy

Reviewer #1: No

---

## [Editor Report · Acceptance letter]

PONE-D-25-22988R1

PLOS ONE

Dear Dr. LeBlanc,

I'm pleased to inform you that your manuscript has been deemed suitable for publication in PLOS ONE. Congratulations! Your manuscript is now being handed over to our production team.

Kind regards,

on behalf of

Dr. Carmelo Caldarella

Academic Editor

PLOS ONE